# Small Molecules Targeting the Specific Domains of Histone-Mark Readers in Cancer Therapy

**DOI:** 10.3390/molecules25030578

**Published:** 2020-01-29

**Authors:** Huihui Zhu, Tao Wei, Yong Cai, Jingji Jin

**Affiliations:** 1School of Life Sciences, Jilin University, Changchun 130012, China; zhuhh15@mails.jlu.edu.cn (H.Z.); taowei16@mails.jlu.edu.cn (T.W.); caiyong62@jlu.edu.cn (Y.C.); 2School of Pharmacy, Changchun University of Chinese Medicine, Changchun 130117, China

**Keywords:** histone marks, histone acetylation, histone methylation, cancer therapy

## Abstract

Epigenetic modifications (or epigenetic tags) on DNA and histones not only alter the chromatin structure, but also provide a recognition platform for subsequent protein recruitment and enable them to acquire executive instructions to carry out specific intracellular biological processes. In cells, different epigenetic-tags on DNA and histones are often recognized by the specific domains in proteins (readers), such as bromodomain (BRD), chromodomain (CHD), plant homeodomain (PHD), Tudor domain, Pro-Trp-Trp-Pro (PWWP) domain and malignant brain tumor (MBT) domain. Recent accumulating data reveal that abnormal intracellular histone modifications (histone marks) caused by tumors can be modulated by small molecule-mediated changes in the activity of the above domains, suggesting that small molecules targeting histone-mark reader domains may be the trend of new anticancer drug development. Here, we summarize the protein domains involved in histone-mark recognition, and introduce recent research findings about small molecules targeting histone-mark readers in cancer therapy.

## 1. Introduction

In eukaryotic cells, histone octamers and DNA together form the basic building block of the chromosome-the nucleosome. The N-terminal tails of the four histones involved in the formation of nucleosomes interact with the acidic patch on the neighboring nucleosomes to form nucleosomal arrays and can be modified by various chromatin-modifying enzymes [1]. Post-translational modifications (PTMs) of the N-terminal tails of histones, such as acetylation, methylation, phosphorylation and ubiquitination, can alter the chromatin structure through affecting the interaction between histones and DNA [2,3]. It is well known that the precise organization of chromatin is important for many intracellular biological processes including gene transcription and recombination [4]. On the other hand, histone marks provide a platform for subsequent protein recruitment [5]. A large number of research data have suggested that different histone marks can be specifically recognized by different proteins (readers). These readers often possess the special domains that recognize and bind to specific histone marks like acetyl-lysines, or identifying methylated lysines or methylated arginine residues. For example, methyl CpG binding domain (MBD) family combined with other factor bind to methylated DNA, which mediates certain genes silencing [6]. Given that abnormal regulation of histone marks is implicated in the occurrence of various diseases including cancer, therefore, it is not difficult to speculate that the functional regulation of the specific domains of histone-mark readers using small molecules may be the trend of new anticancer drug development. In this review, we focus on the histone-mark readers, summarize the recent research findings that link crosstalk between the domain of readers and histone marks, further discuss the effects of small molecules targeting to the specific domain of readers on tumor-epigenetics, and speculate on the potential role that may be relevant to cancer therapy.

## 2. Proteins that Recognize Histone Modification Sites

As the basic structural unit of chromatin, the nucleosome consists of 147 base pairs (bp) of DNA and a histone octamer, which contains two copies each of histones (H2A, H2B, H3 and H4) [7,8]. Histone marks on the N-terminal tails of histones that extended beyond the nucleosome core structure can be regulated by different proteins (or complexes). It is worth mentioning that the histone mark on histone tails is a dynamic and reversible process [9]. Some proteins can add marks to the specific amino acid sites of histones just like writers, while others have roles to remove marks as erasers [10]. In addition, some proteins are recruited at the site of histone marks to obtain instructions for performing functions. However, whether it is to remove the marks or to recruit proteins to a specific modified site, it is usually necessary for the proteins to recognize the specific histone marks just like a reader [11]. Numerous studies have shown that histone-mark readers often recognize marks through the functional domain contained in itself: such as BRD, CHD, PHD and MBT domains. And these special functional domains determine the targeting of readers to different histone marks.

### 2.1. Histone Acetylation Readers

As one of the well-characterized PTMs, histone acetylation is dynamically regulated by histone acetyltransferases (HATs) and histone deacetylases (HDACs) [12,13,14]. Dynamic balance of intracellular acetylation status is involved in many important physiological processes such as gene transcription, cell cycle regulation, DNA replication, DNA damage repair, as well as chromosome remodeling [15]. Known histone-mark readers that can recognize histone acetylation are roughly classified into three categories, including BRD, double PHD finger (DPF) and YEATS domains [16].

#### 2.1.1. BRD-Containing Proteins

BRD-containing proteins are ubiquitously expressed in most tissues and widely present in proteins with multiple catalytic and scaffold functions [17]. Structural analysis have clarified that BRDs contain evolutionarily conserved domains including a left-handed bundle of four α-helices (αZ, αA, αB, αC), linked by ZA and BC loops, which determine binding specificity [18]. Based on sequence or structure similarity, human BRD-containing proteins are divided into eight families [19]. Many proteins with BRD belong to chromatin remodeling or modifying enzymes. As shown in Table 1, BRDs in HATs such as GCN5L2, PCAF, CREBBP act as a protein-protein interaction module that specifically recognize and bind to acetylated histones like H4K8ac, H4K16ac, H4K20ac, H3K14ac, and H3K36ac, thereby affecting subsequent intracellular biological functions [20,21,22]. While the MLL and ASH1L belong to methyltansferases (HMTs), that regulate transcription of genes by targeting H3K4me and H3K36me, respectively [23,24]. It has been reported that MLL-like proteins may target histone H4K16ac. For example, the MOF/NSL HAT complex, which can acetylate histone H4K16ac/K5ac/K8ac, functions in promoting histone H3K4me2 activity by MLL/SET complexes in a unidirectional manner [25], indicating that the MLL/SET complexes may only be recruited in H4K16ac/K5ac/K8ac enriched chromatin.

It is generally believed that acetylation of histones often causes loosened chromatin structure, increasing the accessibility of nucleosomal DNA to transcription factors, thereby activating gene transcription [60]. Just like BRD2 and BRD3 prefer to bind to the chromatin associated with active gene transcription, in which is enriched H4K5ac/K12ac and H3K14ac in human embryonic kidney 293 cells [28]. The BRD family VII member TAF1 facilitates the activation of gene transcription through specific binding to the acetylated histone H4K5/K8/K12/K16 [54]. Compared with TAF1, the BRD family IV member ATAD2 shows multiple functions: (i) Like other BRDs, ATAD2 specifically interacts with acetylated histone H3K14 and diacetylated H4K5/K12 [38,39]; (ii) Assists heterochromatin reassembly during replication through competitively inhibiting the histone deacetylase HDAC1 [38]; (iii) Acting as an E2F co-activator plays a role in E2F-dependent gene activation [39]. Sometimes, BRD proteins may direct the recruitment of chromatin remodeling complex. For example, BRPF1 recognizes acetylated histones (H2AK5ac, H3K14ac, H4K5ac/K8ac/ K12ac) and directs the recruitment of MOZ HAT complex to chromatin [43].

In cells, acetylation of histones often provides a platform for the recruitment of subsequent chromatin remodeling enzymes and collectively performs intracellular biological functions. For instance, BRD4 not only interacts with acetylated histone H3K14ac, H4K5ac/K8ac/K12ac/K16ac [18], but also mediates EP300 to facilitate H3K27ac and H3K56ac at pluripotent genes, and this interaction recruits chromatin remodeler Brg1 to change chromatin structure in embryonic stem cells [29]. In other case, BAZ1B can selectively recognize H2BK12ac, H3K14ac and H4K16ac, and this interaction is indispensable for ligand-induced transrepression of the 1α (OH)ase gene [31]. Furthermore, EP300 and CREBBP share several conserved regions including BRD, HAT domain and cysteine-histidine-rich regions, and function as transcriptional co-activator, playing a critical role in cell proliferation, cell cycle, as well as tumorigenesis [61]. Some BRD proteins can simultaneously recognize acetylation and methylation on histones. ZMYND8 is a good example. Based on the published data, ZMYND8 recognizes acetylated and methylated histones through its PHD/BRD/PWW reader cassette, thereby functions in assembling transcriptional complexes to recruit to DNA-damaged sites [56]. Another example, BRD family I protein BPTF with both BRD and PHD domains can recognize histone H4K16ac and H3K4me3, results in formation of a specific tran- histone modification pattern at mononuleosome level [27]. In addition, the BRD of BAZ2A, a member of BRD family V, recognizes and binds to acetylated histone H4K16, which further recruit NoRC (nucleolar remodeling complex) to trigger deacetylation of histone H4 at K5/K8/K12 sites and NoRC-mediated heterochromatin formation [45].

Some tripartite motif-containing proteins such as TRIM24, TRIM 28, TRIM33 and TRIM66 are members of the human TRIM family also possess the functions in recognizing acetylated histones. For example, TRIM 24 interacts with histone H3K23ac to drive EGFR-mediated tumor growth [51]. TRIM33, a PHD finger-BRD containing protein, determines the binding of TRIM33 to acetylated histone H3K18/K23 while activating E3 ubiquitin ligase activity, and further inhibiting gene transcription [52]. It is worth noting that PBRM1 (Polybromo-1) as a subunit of the Polybromo, Brg1-associated factors (PBAF) chromatin remodeling complex shows high-affinity with acetylated histone H3 at lysine 4, 9, 14 and 23 sites [58]. During mitosis, PBRM1 localizes PBAF to kinetochores through the direct interaction between its six tandem BRDs and acetylated histones [58]. Furthermore, SMARCA2 and SMARCA4 as members of the SWI/SNF chromatin remodeling family interact with DNA and H3K14ac simultaneously, thereby play a role in regulating BAF activity [59].

#### 2.1.2. PHD Finger Proteins

The PHD finger contains a zinc-binding motif (50–80 amino acid residues) that appears in many chromatin-associated proteins [62]. Compared with the BRD-containing proteins, recognition ability of the PHD finger proteins is more flexible, which recognizes acetylated or unacetylated and methylated histones. For example, the PHD6 finger of MLL4 (KMT2D) and the PHD7 finger of paralogous MLL3 (KMT2C) selective target to H4K16ac and provide a direct functional association between MLL3/4 and MOF [63]. MOZ (KAT6A), MORF (KAT6B), DPF1, DPF2 and DPF3 are important DPF proteins that can bind to acetylated lysines [64]. Take MOZ as an example, it can localize to the promoter region of the HOXA9 gene, thereby facilitating H3K14ac around the HOXA9 promoter region and HOXA9 gene transcription [65]. Similarly, MORF binds to the N-terminal tail of histone H3 and shows the preference for K9ac and K14ac [66]. Moreover, fluorescence microscopy and co-immunoprecipitation experiments showed that the PHD1/2 fingers in MORF are required for interaction with H3K14ac in vivo and localization to chromatin forming nuclear speckles [66]. Members of neuron-specific chromatin remodeling complex (BAF) such as DPF2 and DPF3 have been identified to be involved in cancer and embryonic development, respectively. Interestingly, each subunit has a different ability to recognize specific lysine site acetylated histones. DPF2 represses myeloid differentiation in MOLM-13 cells through binding to H3K14ac and H4K16ac via bipartite binding pockets [67]. However, DPF3b interacts with multiple acetylated histones including H3K9ac/K14ac, H4K5ac/K8ac/K12ac/K16ac, and globally co-occurs with histone modifications and BRG1 genomic binding sites, suggesting that DPF3b may act as an anchor between modified histones and the BAF complex [68].

#### 2.1.3. YEATS Proteins

The YEATS family comprises five proteins, Yaf9, ENL, AF9, Taf14, and Sas5. From yeast to human, the YEATS domain is evolutionarily conserved [69]. There are three YEATS domain-containing proteins in *S. cerevisiae* and four in humans. Members of the YEATS family are found to be related to chromatin-remodeling, histone modification, transcription regulation, and DNA repair [70]. In support of the functional importance, dysfunction of YEATS family proteins is often associated with human disease, even cancer. A case in point is AF9, which is the most frequent fusion partner of human MLL proteins caused by chromosome translocations in acute myeloid leukemia (AML) [71]. The recognition of H3K9ac by AF9 is required to recruit DOT1L to chromatin and subsequently deposit H3K79 methylation on target genes to activate transcription [72]. ENL is another YEATS protein relevant to AML. It co-localizes with H3K9ac and H3K27ac on the promoters of actively transcribed genes associated with leukemia, disrupting the linkage between ENL and histone acetylation, decreasing RNA polymerase II recruitment to ENL target genes, thereby results in the suppression of oncogenic gene expression [73]. Besides, the oncogenic gene GAS41, is frequently amplified in human gliomas and preferentially binds to diacetylated histone H3 peptides (H3K18acK27ac) over mono-acetylated histone H3 through a bivalent binding mode [74]. Except for human YEATS family proteins, *yeast* Taf14 contributes to the recognition of histone acetylation. Importantly, block the Taf14-H3K9ac interaction greatly impairs transcriptional regulation coordinated with Gcn5 and sensitizes *S. cerevisiae* cells to DNA damage [75]. Interestingly, histone chaperone Asf1 in *yeast* is a structural homolog of the Yaf9, and recognizes H3K14ac. However, due to the lack of sequence similarity, it is not generally considered a YEATS protein [76].

### 2.2. Histone Methylation Readers

Another well-characterized PTM is a reversible histone methylation (mono-, di- tri-) which is controlled by histone methyltransferases and histone demethylases. The canonical methylated sites contain six lysine residues of histone H3 (K4, K9, K26, K27, K36 and K79), K20 of histone H4 and K26 of histone H1 [77]. These modifications on histone tails mediate widely biological processes, especially gene transcription and DNA damage response. In the past decades, a set of specific-domain containing methyl-lysine mark readers were identified, such as CHD, PHD finger, Tudor, PWWP motif, MBT and WD40 repeat (WDR) domains (Figure 1).

#### 2.2.1. Chromodomain Proteins

The chromodomain (chromatin-organization-modifier domains) has been identified as a 30–70 amino acid residue protein module with a three-stranded β-sheet and an adjacent helix [78]. Chromodomain proteins form or maintain the condensed chromatin structures, and have been involved in transcriptional repression and genome stability [79]. Depending on the functional domain, chromodomain proteins are classified into the heterochromatin (HP1)/polycomb (Pc) family, chromo-ATPase/helicase-DNA-binding (CHD) family, chromobarrel domain family, and the chromodomain Y chromosome (CDY) family [80].

Human HP1 homologs (CBX1, -3, -5) and the Pc homologs (CBX2, -4, -6, -7, -8) are well known as the CBX proteins. The structural and biophysical differences between human HP1 and Pc chromodomains have been uncovered that HP1 homologs possess a large electronegative peptide binding surface, whereas Pc homologs have a much more hydrophobic surface [81]. With the conserved structures, human HP1 family proteins play an important role in heterochromatin packaging and gene silencing such as Chp1, Chp2 and Swi6 in *yeast*, and show a strong preference for H3K9me mark [82,83]. Human Pc homologs exhibit a wide range of affinities for both H3K9me3 and H3K27me3 without a distinct preference [84]. Besides methylated histone lysine binding, certain Pc homologs are capable to bind to nucleic acids. For example, the CBX7-RNA interaction partly regulates the association of CBX7 with H3K9me3 and H3K27me3, and particularly impacts the inactive X chromosome [85].

The structure of chromobarrel domain is similar to that of the HP1/Pc chromodomain, but the chromobarrel domain has an extra β-sheet consisting of two additional strands [79]. *Drosophila* Msl3, human MRG15 and *yeast* Eaf3 are homologs with chromobarrel domain. Msl3, a subunit of male-specific lethal (MSL) complex, is required for the dosage compensation of X-linked genes, and specifically binds to the nucleosomes where is enriched with H3K36me3 or H4K20me1 [86,87], suggesting the methyl-lysine mediated DNA accessibility. Human MRG15 as an adaptor module bind to H3K36me2/3 peptides in a mode different from the canonical peptide binding mode in the HP1/Pc chromodomains [88]. *Yeast* Eaf3 protein is a member of the NuA4 histone acetylase and Rpd3 histone deacetylase complexes. Research data revealed that Eaf3 interacts with three states of methylated H3K36 (me1/2/3) and H3K4me3 marks. Binding of Eaf3 to H3K36me results in the preferential association of the Rpd3 complex, and further inhibiting transcriptional initiation within mRNA coding regions [89]. Another family with chromodomains is the CHD family that contains two N-terminal tandem chromodomains and C-terminal helicase domains. Human CHD1 chromodomains possess secondary structures that are similar to HP1 family chromodomains [90], however, it shows different selectivity for H3K4me1/2/3. In HeLa cells, the interaction of CHD1 with H3K4me3 promotes the recruitment of transcriptional post-initiation and pre-mRNA splicing factors [91]. But in *yeast*, Chd1 does not bind to methylated H3K4 [92], suggesting the different preferences of CHD1 for histone marks in different species. In addition, human CHD7 directly binds to H3K4me which correlates with enhancer mediated transcription during development [93].

Human CDY family includes CDY, CDYL, and CDYL2. Recent reports confirm that the CDY is closely related to male infertility, which may connect to the interaction between CDY chromodomain and the H3K9me2/3 [94,95]. Multimerization of CDYL1b can read H3K9me3, and its association with H3K9me3 is indispensable for the localization of CDYL1b to heterochromatin [96]. But mouse homolous Cdyl binds to not only H3K9me2/3 but also H3K27me3. Furthermore, the combination of H3K9me2 and H3K27me3 recruits Cdyl to Xi, providing a specific binding platform, and facilitates propagation of the H3K9me2 modification by anchoring G9a in mESCs [97].

#### 2.2.2. PHD Finger Proteins

PTMs on histone H3 N- and H4 C-terminal tails provide a perfect basis for histone PTM crosstalk. Despite the PHD fingers can be grouped into several major sub-families according to their binding targets, the PHD-proteins often display diverse histone PTM crosstalk sensitivity. Recent reports describe several examples. In one case, TAF3 as a member of the basal transcription complex (TFIID) directly binds to H3K4me3 and functions as a transcriptional coactivator in a PHD finger dependent manner, and further recruiting TFIID to the same histone marks via the TAF3 PHD finger [98]. Interestingly, asymmetric H3R2me2 can specifically inhibit TFIID binding to the tri-methylated H3K4, in contrast, H3K9ac and H3K14ac promote the interaction between TFIID and H3K4me3 [98]. In lung cancer cells, the interaction between PHF20 and H3K4me2 was disrupted by mutating PHD finger of PHF20, it further reduced MOF mediated H4K16ac targeting gene activation, as well as the preference of PHF20-Tudor2 for p53 dimethylation, suggesting the involvement of PHD finger in anti-cancer processes [99]. In another case, the connection of MLL5 PHD finger with H3K4me3 recruit MLL5 to actively transcribed genes and this association can be disrupted by phosphorylating H3T3 and H3T6, thereby leading to releasing MLL5 from chromatin during mitosis [100].

Histone H3K4me is the major modification recognized by PHD fingers and it mediates various intracellular functions, whereas, minority PHD fingers have been found to bear towards H3K9me3 and H3K36me. Some PHD fingers are evolved with fine-tuned residue composition which integrated or paired with other reader domains, show various readerships. For example, a critical epigenetic remodeler UHRF1 contains multiple domains including Tandem Tudor Domain (TTD)-, PHD-, SRA (SET and RING associated)-, ubiquitin-like- and RING- domains. The PHD finger of UHRF1 preferentially interacts with unmodified H3 N-terminal tail, whereas the TTD cooperates with the PHD finger to specifically bind to H3K9me3 mark during the cell cycle in mammalian cells [101]. Another multi-domain containing protein CHD4 contains two PHD fingers, two chromodomains, and a nucleosome remodeling ATPase domain. Among them, PHD1 recognizes unmodified H3K4, H3K4me3, and H3K9me3, while PHD2 is easier to affinity with unmodified H3K4 and H3K9me3 [102]. Likewise, both PHF8 and JHDM1D are demethylases that can harbor a PHD binding to H3K4me3. Interestingly, the contact of PHF8-H3K4me3 enhances the demethylation activity on H3K9me2, in contrast, JHDM1D-H3K4me3 interaction inhibits the demethylation activity on H3K9me2 [103].

#### 2.2.3. Tudor Domain Proteins

Tudor domain contains approximately sixty amino acids that are composed of 5 stranded β-barrel fold with one or two helices packed against the parallel β-sheet [104]. Tudor domain- containing proteins binding to methylated lysine or arginine residues function as molecular adaptors in differentiation, cell division, gametogenesis, and genome stability. Both PHF1 and PHF19 contain a single Tudor motif, a C-terminal chromo-like domain, and two PHD fingers. Based on recent research data, both PHF1 and PHF19 can selectively read H3K36me2/3 modification, and the interaction between PHF proteins and H3K36me inhibits the catalyzation activity of HMT PRC2 (Polycomb repressive complex 2) to facilitate the silencing of transcribed genes in human cancer cells and embryonic stem cells [105,106,107]. In addition, PHF1 regulates the deposition of the repressive H3K27me3 mark and functions as a cofactor in the PARP1- and Ku70-Ku80-initiated DNA damage response [106]. Some proteins contain two five-stranded β-barrels connected by a short (3-5 aa) linker termed TTD [108]. Just like JMJD2 family members JMJD2A, JMJD2B, and JMJD2C possess TTD domain, and are capable of removing tri-/di-methylation marks on H3K9 and H3K36 [109]. Structural analysis further shows that the JMJD2A-TTD recognizes H3K4me3 and H4K20me3 through which forms a bilobal, saddle-shaped structure, with each hybrid lobe resembling a canonical Tudor fold, indicating a role in chromatin-targeting [110,111]. In cells, TTD in JMDJ2A together with 53BP1 forms two independent folded structures, interacting with H4K20me2, thereby contributing to the accumulation of 53BP1 to sites of IR-induced DNA DSBs [112,113].

#### 2.2.4. PWWP Motif Proteins

The PWWP domain, with a conserved Pro-Trp-Trp-Pro motif, consists of a five-stranded β-sheet packed against a variable helical bundle [108]. According to the literature, PWWP domain proteins possess methyl-lysine recognition activity like H3K36me3, H3K79me3, H4K20me3 and other histone modifications. Fluorescence polarization assay confirms that hepatoma-derived growth factor (HDGF) with highly conserved PWWP motif binds to methylated H3K36, H3K79 and H4K20 [114]. Another example is that the transcription elongation factor TbTFIIS2-2 can also read H4K17me3 and H3K32me3 marks via the conserved aromatic cage of the PWWP domain in its N-terminal [115]. Both PWWP domain containing proteins LEDGF (chromatin-associated protein PSIP1) and ZMYND11 selectively recognizes H3K36me3 and functions in alternative or RNA splicing [116,117]. In addition, LEDGF is essential for MLL-rearranged leukemia [118], and plays a key role in the HIV life cycle [119]. Furthermore, LEDGF interacts with other factors through unstructured regions that can be blocked by inhibiting serine/threonine kinases [120]. Hence it is a particularly interesting H3K36me3 reader for many diseases, and some of its interactions can be targeted by repurposing non-chromatin drugs. In *yeast*, a PWWP domain-containing protein Pdp3 recruits NuA3 HAT to chromatin through Pdp3 binding to H3K36me3, and coordinating transcriptional elongation at coding region [121,122]. However, a mutation in the PWWP motif of DNA methyltransferase DNMT3A abrogates its interaction with H3K36me2/3, and results in postnatal growth deficiency in mice [123], suggesting the importance of PWWP domain in biological functions. Moreover, the PWWP domain containing protein Pdp1 binds to the methylated histone H4K20 and DNA simultaneously, thereby regulates the catalysis activity of Set9 enzyme to regulate the methylation status of H4K20 by specifically recognizing H4K20me mark [124]. Although the existence of H4K20me3 does not impact on Pdp1-PWWP domain binding to DNA, the DNA binding activity may provide the energy to the PWWP domain to recognize the methylated-nucleosome. This coordinative function is essential for intracellular biological processes. Also, in vivo experiments have verified that the hepatocellular carcinoma pathogenesis associated protein HRP3 recognizes H3K36me2/3 marks and dsDNA, simultaneously, and plays a key role in recruiting HRP3 to the chromatin [125], suggesting that the PWWP binds histone and DNA with different binding pockets.

#### 2.2.5. WDR and MBT Domain Proteins

The WDR domain is typically a seven-bladed β-propeller domain with an overall doughnut shape which is involved in a broad range of cellular pathways including ubiquitin- and immune- related pathways, DNA damage response, cell cycle and chromatin remodeling [126]. The WDR domain-containing protein EED as a subunit of the PRC2 complex binds to H3K27me3 to enhance H3K27me3 activity of the catalytic subunit EZH2 [127]. Joanna et al. found that WDR5 directly and specifically associated with H3K4me2/3, and this interaction is essential for global H3K4me3 and HOX gene activation in human cells [128]. Furthermore, the structural analysis revealed that WDR5 cannot read out the methylation state of H3K4 directly. Instead, it presents the H3K4 side chain for further methylation by SET1 family proteins [129].

The MBT domain is highly conserved from *C. elegans* to humans that selectively recognizes and binds to methylated histones [130,131]. *Drosophila* Sfmbt1, a subunit of histone demethylase LSD1 complex, contains four MBT repeats and functions in HOX gene silencing through binding to H3K9me1/2 and H4K20me1/2 [132]. Unlike the protein in *Drosophila*, human SFMBT1 recognizes histone H3K4me2/3 and these modifications are required for LSD1 recruitment to chromatin, H3K4 demethylation, and TGFβ-induced epithelial to mesenchymal transition [133]. In *C. elegans*, the MBT-containing protein LIN-61 specifically interacts with H3K9me2/3 and functions in the synthetic multivulval (synMuv) pathway of vulva development [134].

## 3. Domain-Specific Inhibitors that Target Histone-Mark Readers

Imbalanced global histone marks in cells have been involved in initiating events of cancer by aberrant regulating oncogenes and/or tumor suppressors [135]. Given that the histone marks on histone residues are the reversible processes, the roles of histone-mark readers in restoring the abnormal histone modifications to normal can be speculated. Therefore, small molecules that target the specific domains of -histone-mark readers have the potential to reverse abnormal gene expression during tumorigenesis, thus providing a new way to develop cancer therapeutic drugs. Previous literature well reviewed the chemical biology tools including selective PROTAC E3 ubiquitin ligase degraders, degrons, fluorescent ligands, dimerizers inhibitors, and other drugs. These small molecules can control the activities of deregulated epigenetic modulators, thereby changing intracellular biological processes [136]. In this section, we summarized domain-specific inhibitors that targeting histone-mark readers. The good news is that some small-molecule compounds have already begun trial use in the clinic.

### 3.1. BET/BRD Inhibitors

BRD domains have been identified in nuclear proteins including HATs, HMTs, chromatin remodeling enzymes and transcriptional co-activators. As mentioned earlier, the BRDs as the readers of acetyl marks on histone tails play a key role in gene transcriptional activation through targeting acetylated chromatin. Because of this, people have long set their sights on the development of small molecules that target BRDs in an attempt to achieve the purpose of treating cancer. According to the structural features of BRDs, the ‘WPF shelf ’and ‘gatekeeper’ residue have been considered as two important specificity determinants of BRD inhibitors. The two residues of ‘WPF shelf’ change in hydrophobicity, size, or effects on backbone conformational properties, while ‘gatekeeper’ residue determines the shape of the entrance into the acetylation binding site [137]. These differences make the two areas be explored to achieve selectivity depending on the desired targets. Currently, several small molecules (BRDs inhibitors) that target the acetyl-binding pockets of the BRD or BRD-extraterminal proteins (BETs) have been developed. Some inhibitors like benzodiazepines and quinolines have been shown to have significant anti-proliferative activity against a variety of hematologic- and solid- tumors [138]. For example, the benzodiazepine JQ1 as a cell permeable small molecule can competitively bind to BRDs with high potency and specificity. A large number of studies have showed the efficacy of JQ1 in hematological malignancies and in a variety of solid tumors including glioblastoma neuroblastoma, breast cancer, pancreatic cancer, and lung cancer by repressing of c-MYC, FOSL1, or other transcriptional targets [139,140,141,142,143]. The quinolone class of BET inhibitors I-BET151 and I-BET-762 induce apoptosis and cell cycle arrest through down-regulation of MYC and up-regulation of HEXIM1 [144]. Recently, more BET inhibitors have been developed such as MK-8628 and PF-1. In mouse glioblastoma cells, MK-8628 (OTX015) showed much higher anti-proliferative effect than JQ1 [145]. PF-1 as an acetyl-lysine (Kac) mimetic inhibitor occupies the acetyl-lysine binding site in BRD2 and BRD4, thereby blocking the interaction of BET bromodomains with acetylated histone tails, and results in arrest of the cell cycle in G1, down-regulation of MYC, Aurora B kinase, and apoptosis induction [146]. There are similar BET inhibitors such as BMS-986158 and PLX-51107 which can also bind to the acetyl-lysine binding site in the BRD of BET proteins and disrupt the interaction between BET proteins and acetylated histones, thereby preventing the expression of certain growth-promoting genes, resulting in an inhibition of tumor cell growth [147]. Recent research data reveal that bromosporine, an innovative BET inhibitor, acts as a broad-spectrum BRD inhibitor has been confirmed to enhance 5-FU effect in colorectal cells under laboratory studies [148].

The most delighted thing is that a number of BET inhibitors such as RVX-208 (RVX00022), I-BET762 (GSK525762), FT-1101, CPI-0610, BAY1238097, INCB054329, TEN-010, GSK2820151, ZEN003694, BMS-986158, BI 894999, ABBV-075, GS-5829, PLX51107 and OTX015 (MK-8628) have been under clinical trials to investigate numerous cancer types or other diseases (Table 2). atherosclerosis, coronary syndromes and Alzheimer disease [149,150,151,152,153].

For example, ZEN003694, GS-5829 and OTX015 are currently underway in clinical trials to treat castration-resistant prostate cancer. Recent developed BI 894999, FT-1101 and CPI-0610 are undergoing clinical trials in hematological cancers. BAY1238097, I-BET762, INCB054329, BMS-986158, ABBV-075 and PLX51107 are being clinically tested for several types of cancer. While the TEN-010 and GSK2820151 are being tested to treat solid tumors or hematological cancers, respectively. Some BET inhibitors are also used to treat human diseases other than cancer. RVX-208 (RVX00022), a BET inhibitor with preferred binding to the second bromodomains (BD2s) of BRD2 and BRD3, is currently being investigated in several clinical trials for cardiovascular diseases.

In recent years, with the deepening of epigenetics research, it has been found that in many cases the recognition of histone marks needs to be accomplished through non-BET functional domains. Given that many proteins with BRD belong to chromatin remodeling or modifying enzymes, the inhibitors of non-BET may achieve the purpose of treating cancer through correcting the epigenetic modification abnormalities caused by cancer. Several small molecules (I-BRD9, BI-7273/BI-9564, BI-7271/BI-7189, PFI-3, MS2126/MS7972, SGC-CBP30, PF-CBP1 and I-CBP112) as the BRD inhibitors have been reported [154]. Although the above small molecules are rarely used in clinical trials, some compounds have been clarified to have effects on several cancer cells in laboratory studies. For example, I-BRD9 may be used to identify downstream genes of BRD9 involved in oncology and immune response pathways in Kasumi-1 cells [155]. In addition, compounds BI-7273/BI-9564/ BI-7271/BI-7189 as inhibitors of BRD9 selectively suppress the proliferation of mouse and human AML cells [156,157]. Some non-BET inhibitors like MS2126/MS7972 target the transcriptional co-activators CREBBP and EP300, and result in the low level of p53 phosphorylation at Ser15 and acetylation at Lys382 via inhibiting the interaction of CREBBP and P53 in DNA damage response [158]. Subsequently, three chemical compounds including SGC-CBP30, PF-CBP1, and I-CBP112 were generated with higher potency and selectivity, showing anti-cancer effects on multiple myeloma and prostate cancer [159,160]. Fortunately, a p300/CBP BRD inhibitor CCS1477 is underway in clinical trials for treat metastatic prostate cancer and other solid tumours [161].

### 3.2. PHD Inhibitors

As mentioned earlier, the PHD finger is a versatile reader of histone marks that can selectively recognize unmethylated, methylated, and acetylated lysine residues [162]. Compared with BRD inhibitors, less attention has been focused on other functional domains such as PHD, and most of them are still in the preclinical stage. Based on research data, there are two modes of histone-mark recognition of PHD fingers, the cavity insertion and surface groove [163], and these two modes are commonly seen in readers with PHD fingers and chromodomains, as well as MBT domains and the Tudor domain 53BP1, respectively [164]. Because of the high conservation of the binding areas, the usage of secondary pockets is the strategy of choice to achieve the specificity [165]. Therefore, the development of PHD inhibitors has focused on the molecules that can selectively bind to cavity insertion and surface groove to attenuate or eliminate the ability of PHD to recognize the histone marks. For example, small molecule macrocyclic calixarenes can disrupt binding of ING2 PHD to methylated H3K4 in vivo and in vitro [166]. Furthermore, small molecules including disulfiram, amiodarone and tegaserod were identified as inhibitors of the interaction between JARID1A PHD3 and H3K4me3 using Halo Tag assay [167]. Shortly, fragment-based NMR screening approach discovered a small molecule benzimidazole that can selectively dock into the methylated H3K4 and displace the native H3K4me peptide from PHD finger of the Pygo-BCL9 chromatin reader [168].

### 3.3. BMT and Tudor Inhibitors

Chromatin-binding domain BMT recognizes methylated histone H3K4. Usually, the narrow aromatic cage pocket of MBT domain only allows lysine Kme1 and Kme2 bound, and results in repression of gene expression that involved in several disease states [130,169]. Although there is currently little known on the therapeutic effects of MBT domain inhibitors on clinical diseases, several compounds have been developed as the potent and selective inhibitors for MBT-domain proteins. For example, UNC669 and UNC926 are reported as the first inhibitors for L3MBTL1 [170,171], while the UNC1215, UNC1079, UNC1679, UNC56 and UNC2533 are discovered as the inhibitors for L3MBTL3 [164,172]. Some inhibitors like UNC1215 have revealed a regulatory function on apoptosis related protein BCLAF1 during DNA damage repair through interacting with L3MBTL3 [172] Further research found that some small UNC compounds can also be used as inhibitors of Tudor domain. It has been known that proteins with Tudor domains such as UHRF1, JMJD2A and 53BP1 play a critical role in cells as ‘methyl-mark’ readers. A small molecule UNC2170 inhibits 53BP1 binding at the interface of two Tudor domains of a 53BP1 dimer, showing cellular activity by suppressing class switch recombination [173]. Recent research shows that the clinically used iron chelator deferasirox potently inhibit JMJD2A, leading to a high level of histone trimethylation and inhibition of cancer cell growth [174]. What is more, UHRF1 inhibitors as an alternative of DNA demethylation agents can inhibit DNA methylation in cancer therapies [175].

### 3.4. CHD Inhibitors

CHD domains are a family of methyl-mark readers. HP1 and Pc family that typically bind to H3K9me3 and H3K27me3, respectively, are the most thoroughly studied CHD families. As mentioned earlier, the human paralogs of HP1 and Pc (chromobox, CBX proteins) are divided into eight CBX proteins, with CBX1/3/5 belonging to the HP1 family and CBX2/4/6/7/8 to the polycomb family. According to the cell experiments, each CBX protein has distinct roles at different stages of various aggressive cancers [176,177]. Recently, CBX7 and CBX8 have emerged as a potential therapeutic target for lymphoma, hepatocellular carcinoma, leukemia and breast cancer [177,178,179,180]. Therefore, small molecule CHD inhibitors target CBX7 and CBX8 are studied. MS37452 was developed as a selective ligand for CHD of CBX7. In PC3 prostate cancer cells, MS37452 reduced CBX7 occupancy at INK4A/ARE, showing transcriptional de-repression of gene products p14/ARE and p16/INK4a [181]. Another small molecule is the SW2_110A, a selective and cell-permeable inhibitor of the CBX8. It has been reported that SW2_110A specifically inhibits the proliferation of THP1 leukemia cells driven by MLL-AF9 translocation, resulting in significant decreased expression of MLL-AF9 target genes [182]. The small molecule UCN3866 was confirmed as a potent antagonist of the methyl-tag readers such as polycomb CBX and CDY families of CHDs, and can inhibit PC3 prostate cancer cell proliferation through down-regulating transcription of the INK4a/ARE locus [183].

## 4. Conclusions and Perspectives

It is clear that histone-mark reader proteins play an important role in basic biological processes in cells. Importantly, the different functional domains on these readers recognize different histone marks based on their specific binding pocket. After years of efforts, although there are many small molecule compounds developed to disrupt the interaction between specific domains and epigenetic-tags, few compounds can be used in clinical trial to treat cancers. Given that cancers may be caused by multiple factors such as the changes of cell microenvironment, imbalanced intracellular histone marks, the dysfunction of histone-mark readers and dis-coordinative function between different epigenetic mechanisms, etc. Therefore, the development of effective, multi-target small molecule compounds may be the key to uncover the cross-talk between histone-mark readers and histones, along with non-histone proteins. It will continue to attract more attention of researchers to develop better inhibitors in cancer therapy.

## Figures and Tables

**Figure 1 molecules-25-00578-f001:**
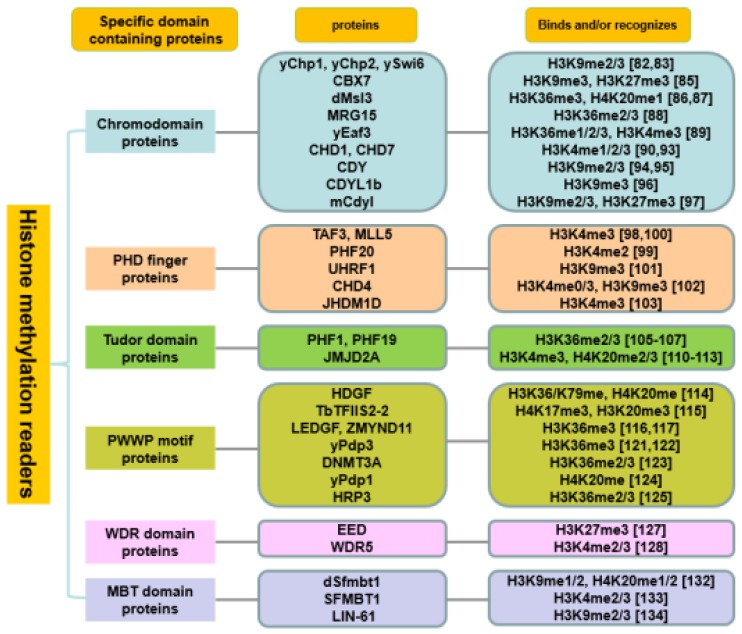
Histone methylation readers.

**Table 1 molecules-25-00578-t001:** Human bromodomain family and its features.

Groups	Proteins	Name	BRDs	Alias	Functions and Recognize Histones
**BRD I**	BAZ1A	Bromodomain adjacent to zinc finger domain, 1A	1	ACF1, WALp1, WCRF180	Chromatin remodeling factor [26]
	BPTF	Fetal Alzheimer antigen	1	FALZ, FAC1	Recognizes H4K16ac and H3K4me3 [27]
	CECR2	Cat eye syndrome chromosome region, candidate 2	1	KIAA1740	Recognizes/binds to acetylated histones [18]
	GCN5L2	General control of amino acid synthesis 5-like 2	1	GCN5, KAT2A, STAF97, PCAF-B	HAT, interact with EP300/CREBBP and ADA2; Binds to H4K16ac [20,22]
	PCAF	P300/CBP-associated factor	1	CREBBP-associated factor, KAT2B	HAT, promotes transcriptional activation; Targets to H4K8ac, H3K14ac and H3K36ac [21,22]
**BRD II (BET)**	BRD2	Bromodomain-containing protein 2	2	FSH, RING3	Associated with acetylated chromatin during mitosis; Binds to the H4K5ac/K12ac and H3K14ac [28]
	BRD3	Bromodomain-containing protein 3	2	ORFX, RING3L	Transcriptional regulator; Binds to the H4K5ac/K12ac and H3K14ac [28]
	BRD4	Bromodomain-containing protein 4	2	CAP, MCAP, HUNK1	Interacts with acetylated H3K14 and H4K5/K8/K12/K16; Rerulates H3K27ac and H3K56ac [18,29]
	BRDT	Bromodomain-containing protein, testis specific	2	BRD6	Chromatin remodeling factor; Recognizes H4K5ac/K8ac [30]
**BRD III**	BAZ1B	Bromodomain adjacent to zinc finger domain, 1B	1	WSTF, WBSCR9	Chromatin remodeling factor, transcriptional regulator; Recognizes H2BK12ac, H3K14ac and H4K16ac [31]
	BRD8B	Bromodomain-containing protein 8 B	2	SMAP, SMAP2	Transcriptional regulator [32]
	BRWD3	Bromodomain-containing protein disrupted in leukemia	2	BRODL	Associated with translocations in patients with B-cell chronic lymphocytic leukemia [33]
	CREBBP	CREB Binding Protein	1	CBP, KAT3A	HAT; Binds to H4K20ac [22]
	EP300	E1A-binding protein p300	1	p300, KAT3B	HAT, acetylate H3K122/K27 and non-histone proteins [34,35]
	PHIP	Pleckstrin homology domain-interacting protein	2	DR11, WDR11, SRT1, HH14, BRWD2	Binds to the insulin receptor substrate 1; Regulates growth and survival of pancreatic beta cells [36]
	WDR9	WD repeat domain 9	2	BRWD1	Chromatin remodeling factor [37]
**BRD IV**	ATAD2	Two AAA domain containing protein	1	ANCCA	Transcriptional regulator; Binds to H3K14ac, H4K5ac/K12ac [38,39]
	ATAD2B	KIAA1240 protein	1	KIAA1240	Binds to acetylated chromatin [gene card]
	BRD1	Bromodomain-containing protein 1	1	BRPF2	A subunit of the MOZ/MORF HAT complex, regulates H3K14ac [40]
	BRD7	Bromodomain-containing protein 7	1	BP75, NAG4, CELTIX1	Transcriptional regulator; Binds to acetylated H3K9/K14, H4K8/K12/K16 [41]
	BRD9	Bromodomain-containing protein 9	1	LAVS3040, PRO9856	Chromatin remodeling; Recognizes H4K5ac/K8ac [42]
	BRPF1	Bromodomain- and PHD finger-containing protein 1A	1	BR140, Peregrin	Transcriptional activator; Recognizes acetylated H2AK5 and H3K14, H4K5/K8/K12 [43]
	BRPF3	Bromodomain- and PHD finger-containing protein, 3	1	KIAA1286	Regulats replication origin activation and histone H3K14 acetylation [44]
**BRD V**	BAZ2A	Bromodomain adjacent to zinc finger domain, 2A	1	TIP5, WALp3	Transcriptional repressor; Interacts with H4K16ac [45]
	BAZ2B	Bromodomain adjacent to zinc finger domain, 2B	1	WALp4	Transcriptional regulator; Binds to H3K14ac [46]
	SP100	Nuclear antigen Sp100	1	Lysp100b	Binds heterochromatin and functions in immunity, and gene regulation [47]
	SP110	Nuclear antigen Sp110 A,	1	IPR1, VODI, IFI41, IFI75	Transcriptional activator [48]
	SP140	SP140 nuclear body protein	1	LYSP100	Associated with multiple sclerosis, Crohn’s disease, chronic lymphocytic [49,50]
	SP140L	SP140 nuclear body protein like	1	SP140L-1 protein	Chromatin binding [gene card]
	TRIM24	Tripartite motif-containing 24	1	TIF1a, PTC6, RNF82	Transcriptional regulator; Recognizes H3K23ac [51]
	TRIM33	Tripartite motif-containing 33 A	1	PTC7, RFG7, TIF1g	Control transcriptional elongation; Recognizes H3K18ac/K23ac [52]
	TRIM66	Tripartite motif-containing 66	1	TIF1d	Transcriptional repressor [53]
**BRD VI**	MLL	Myeloid/lymphoid or mixed lineage leukemia	1	HRX, TRX1, CXXC7, ALL-1	HMT, mediats H3K4me [23,25]
	TRIM28	Tripartite motif-containing 28	1	KAP1, RNF96, TIF1b	Transcriptional regulator; Regulates H3K9ac and H3K14ac [gene card]
**BRD VII**	BRWD3				
	PHIP				
	TAF1	TAF1 RNA polymerase II, TATA box-binding protein (TBP)-associated factor	2	TAFII250, P250, CCG1, TAF2A	Transcription inhibition; Binds to H4K5/K8/K12/K16ac [54]
	TAF1L	TAF1-like RNA polymerase II, TATA box-binding protein (TBP)-associated factor	2	TAF(II)210	Functions as a TBP-associated factor [55]
	WDR9				
	ZMYND8	Zinc Finger MYND-Type Containing 8	1	PRKCBP1, RACK7	Transcriptional regulator; Recognizes H4K5/K8/K12/K16/K20ac H3K9ac, and H3K14ac [56]
	ZMYND11	remodeling factor containing 11	1	BS69, BRAM1, MRD30	Transcriptional repressor [57]
**BRD VIII**	ASH1L	ash1 (absent, small, or homeotic)-like	1	ASH1, KMT2H	HMT, methylates H3K36me2 [24]
	PBRM1	Polybromo 1	6	PB1, BAF180	Chromatin remodeling factor; High-affinity with acetylated histone H3 at lysine 4, 9, 14 and 23 [58]
	SMARCA2	SWI/SNF-related matrix associated actin-dependent regulator of chromatin a 2	1	BRM, SNF2L2	Chromatin remodeling factor, Splicing regulator; Interact with and moderate specificity for H3K14ac [59]
	SMARCA4	SWI/SNF-related matrix associated actin-dependent regulator of chromatin a 4	1	BRG1, SNF2L4, SNF2LB	Chromatin remodeling factor; Interact with and moderate specificity for H3K14ac [59]

**Table 2 molecules-25-00578-t002:** BET inhibitors used in clinical trials (from clinicaltrials.gov as of September 2019).

Compounds	Conditions	Status	Clinical Trials Identifier
ABBV-075	AML; Advanced Cancer; Breast Cancer; Multiple Myeloma; NHL; NSCLC; Prostate Cancer	Completed Phase 1	NCT02391480
BAY1238097	Neoplasms	Terminated Phase 1	NCT02369029
BI 894999	Neoplasms	Recruiting Phase 1	NCT02516553
BMS-986158	Advanced Tumors	Recruiting Phase 1/2	NCT02419417
	Childhood Solid Tumor; Lymphoma; Pediatric Brain Tumor	Recruiting Phase 1	NCT03936465
CPI-0610	AML; Myelofibrosis; MMN; MS	Recruiting Phase 1/2	NCT02158858
	Lymphoma	Completed Phase 1	NCT01949883
	Multiple Myeloma	Completed Phase 1	NCT02157636
	Peripheral Nerve Tumors	Withdrawn Phase 2	NCT02986919
FT-1101	AML; Acute Myelogenous Leukemia; MS; NHL	Completed Phase 1	NCT02543879
GS-5829	Advanced Estrogen Receptor; Positive HER2-Breast Cancer	Terminated Phase 1/2	NCT02983604
	Lymphomas, Solid Tumors	Completed Phase 1	NCT02392611
	Metastatic CRPC	Active, not recruiting Phase 1/2	NCT02607228
GSK2820151	Solid Tumors	Active, not recruiting Phase 1	NCT02630251
GSK525762 (I-BET762)	Advanced and Retractory Solid Tumors; Lympnomas	Active, not recruiting Phase1	NCT03925428
	Carcinoma Midline	Active, not recruiting Phase 1	NCT01587703
	Drug Interactions	Completed Phase 1	NCT02706535
	Neoplasms	Recruiting Phase 2	NCT01943851
	Neoplasms in combination with fulvestrant	Recruiting Phase 2	NCT02964507
	Solid Tumors	Recruiting Phase 1	NCT03150056
	Solid Tumours	Withdrawn Phase 2	NCT03266159
	Solid Tumours	Available	NCT03702036
INCB054329	Hematologic Malignancy; Solid Tumors	Terminated Phase 1/2	NCT02431260
MK-8628 (OTX015)	AML	Active, not recruiting Phase 1	NCT02698189
	AML	Withdrawn Phase 1/2	NCT02303782
	Acute Lymphoblastic Leukemia; AML; Diffuse Large B-cell Lymphoma; Multiple Myeloma	Completed Phase 1	NCT01713582
	Glioblastoma multiforme	Terminated Phase 2	NCT02296476
	CRPC; NUT Midline Carcinoma; NSCLC; Triple Negative Breast Cancer	Terminated Phase 1	NCT02698176
	CRPC; NUT Midline Carcinoma; NSCLC With Rearranged; ALK Gene/Fusion Protein or KRAS Mutation; Pancreatic Ductal Adenocarcinoma;Triple Negative Breast Cancer	Completed Phase 1	NCT02259114
PLX51107	AML; MS; NHL; Solid Tumors	Terminated Phase 1	NCT02683395
	AML; MS; MMN; Myeloproliferative Neoplasm	Not yet recruiting Phase 1	NCT04022785
RVX000222	Fabry Disease	Not yet recruiting Phase 1/2	NCT03228940
	Cardiovascular Diseases; Coronary Artery Disease; Type 2 Diabetes Mellitus	Active, not recruiting Phase3	NCT02586155
TEN-010 (RO6870810)	Advances solid malignancies; Solid Tumors	Completed Phase 1	NCT01987362
	AML; Myelodysplastic Syndromes	Completed Phase 1	NCT02308761
	Multiple Myeloma	Active, not recruiting Phase 1	NCT03068351
ZEN003694	Metastatic CRPC	Completed Phase 1	NCT02705469
	Metastatic CRPC in combination with Enzalutamide	Active, not recruiting Phase 1/2	NCT02711956

AML, Acute Myeloid Leukemia; NSCLC, Non-Small Cell Lung Cancer; NHL, Non-Hodgkins Lymphoma; MS, Myelodysplastic Syndrome; MMN, Myelodysplastic/Myeloproliferative Neoplasm; CRPC, Castrate-resistant Prostate Cancer.

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
