# Peer review of "Small Molecules Targeting the Specific Domains of Histone-Mark Readers in Cancer Therapy"

_molecules, 2020, doi:10.3390/molecules25030578_

Round 1
Reviewer 1 Report
This manuscript is a thorough and comprehensive review of chromatin binding domains, and discusses their roles in human disease and suitability for drug targeting. I generally like it, and believe that it ought to eventually be published, but there are many issues with it currently. While it is generally well written, there are many clumsy English issues, such as singular/plurals mixing, and articles that are not used correctly. While these are generally minor readability issues and the intent of the sentences is often relatively clear, there are very many of these (too many to count), and as a result the text is sometimes difficult to read for a native English speaker. Authors would benefit from a copy editing service to address these small but pervasive grammar issues.
In the introductions and conclusions, "epigenetic tags" uses quotes every time it is written, which is very unusual. It may reflect an attempt at search engine optimization, but whether or not this is true, it feels extraordinarily unnatural and should be removed throughout the text. Additionally, some readers may feel that "epigenetic" is not the best word choice to describe all histone marks, some of which are not heritable and therefore are not truly "epigenetic" (see Ptashne, Current Biology 17(7): R233, 2017). As such, in most instances, "chromatin mark," "histone mark," or "modified histone" (all without quotes when written in the actual text) are more accurate word choices than "epigenetic tag" in this review.
Additional citations
Authors have created a unique and very thorough resource for the community that builds on previous similar work. However, authors should cite previous reviews published in this same journal covering similar topics, such as Cermakova & Hodges, Molecules (2018), as well as highlight for readers the ways in which their current work improves upon the existing published literature.
In section 2.2.4 about PWWP domains, some aspects of LEDGF may also be relevant for this review. LEDGF is essential for MLL-rearranged leukemia (Cermakova et al., Cancer Research 2014), and also plays a key role in the HIV life cycle (Tesina et al., Nature Communications 2015). It also interacts with other factors through unstructured regions that can be blocked by inhibiting serine/threonine kinases (Sharma et al., Proc Natl Acad Sci USA 2018). Hence it is a particularly interesting H3K36me3 reader for many diseases, and some of its interactions can be targeted by repurposing non-chromatin drugs. Additional citations such as those above would add depth to this section.
Minor issues that need correction:
Line 13: In the abstract, authors identify that CHD means "chromosomal domain," yet later in the text, identify it to mean "chromodomain." The usage in the abstract is incorrect, and should be corrected to "chromodomain."
Table 1: The primary gene name for Polybromo 1 is "PBRM1" rather than "PB1"
Line 207: CDH1 should be corrected to CHD1
Line 210: Again, CDH1 should be corrected to CHD1
Line 210: Scare quotes around 'histone marks' should be removed, and this should be performed everywhere such quotes are used throughout the text. The authors are not introducing these terms, as these are widely used vocabulary that do not need such quotes.
Line 225: Quotes should be removed, here and all other places where they are used around similar terms
Lines 220 and 380 are extremely similar and almost duplicate each other, one should be changed
Lines 407 and 414: Is "methy-tag" a typographical error for "methyl tag?" If so, this is not a widely used term. This term should either be corrected, or introduced and defined.
The paper has no figures or illustrations. A conceptual image illustrating the features they describe in the text would be very helpful.
Author Response
Reviewer 1:Comments and Suggestions for Authors
This manuscript is a thorough and comprehensive review of chromatin binding domains, and discusses their roles in human disease and suitability for drug targeting. I generally like it, and believe that it ought to eventually be published, but there are many issues with it currently. While it is generally well written, there are many clumsy English issues, such as singular/plurals mixing, and articles that are not used correctly. While these are generally minor readability issues and the intent of the sentences is often relatively clear, there are very many of these (too many to count), and as a result the text is sometimes difficult to read for a native English speaker. Authors would benefit from a copy editing service to address these small but pervasive grammar issues.
- In the introductions and conclusions, "epigenetic tags" uses quotes every time it is written, which is very unusual. It may reflect an attempt at search engine optimization, but whether or not this is true, it feels extraordinarily unnatural and should be removed throughout the text. Additionally, some readers may feel that "epigenetic" is not the best word choice to describe all histone marks, some of which are not heritable and therefore are not truly "epigenetic" (see Ptashne, Current Biology 17(7): R233, 2017). As such, in most instances, "chromatin mark," "histone mark," or "modified histone" (all without quotes when written in the actual text) are more accurate word choices than "epigenetic tag" in this review.
We agree with reviewer’s comment. The quotes for "epigenetic tags" have been removed throughout the text. Also, in revised manuscript, most "epigenetic tags" are replaced by "histone mark" and above reference was cited as reference [5].
Additional citations
Authors have created a unique and very thorough resource for the community that builds on previous similar work. However, authors should cite previous reviews published in this same journal covering similar topics, such as Cermakova & Hodges, Molecules (2018), as well as highlight for readers the ways in which their current work improves upon the existing published literature.We appreciate the reviewer’s suggestion. In revised manuscript, we cited the above article as reference [136], and related content added in section 3.
In section 2.2.4 about PWWP domains, some aspects of LEDGF may also be relevant for this review. LEDGF is essential for MLL-rearranged leukemia (Cermakova et al., Cancer Research), and also plays a key role in the HIV life cycle (Tesina et al., Nature Communications 2015). It also interacts with other factors through unstructured regions that can be blocked by inhibiting serine/threonine kinases (Sharma et al., Proc Natl Acad Sci USA 2018). Hence it is a particularly interesting H3K36me3 reader for many diseases, and some of its interactions can be targeted by repurposing non-chromatin drugs. Additional citations such as those above would add depth to this section.
We appreciate the reviewer’s suggestion. In section 2.2.4, above contents have been added. The above articles cited as references [118],[119],[120].
Minor issues that need correction:
Line 13: In the abstract, authors identify that CHD means "chromosomal domain," yet later in the text, identify it to mean "chromodomain." The usage in the abstract is incorrect, and should be corrected to "chromodomain."
We appreciate the reviewer’s comments. We have changed "chromosomal domain" in line 13 to "chromodomain".
Table 1: The primary gene name for Polybromo 1 is "PBRM1" rather than "PB1"
According to the reviewer’s suggestion, the Polycomb 1 abbreviation replaced original PB1 with PBRM1.
Line 207: CDH1 should be corrected to CHD1
Line 210: Again, CDH1 should be corrected to CHD1
We have corrected “CDH1” to “CHD1” in line 207 and 210.
Line 210: Scare quotes around 'histone marks' should be removed, and this should be performed everywhere such quotes are used throughout the text. The authors are not introducing these terms, as these are widely used vocabulary that do not need such quotes.
We agree with reviewer’s comment. Scare quotes '---' have been removed throughout the text.
Line 225: Quotes should be removed, here and all other places where they are used around similar terms
We agree with reviewer’s comment. Scare quotes '---' have been removed throughout the text.
Lines 220 and 380 are extremely similar and almost duplicate each other, one should be changed.
Sorry we did not find the relevant content mentioned by the reviewer.
Lines 407 and 414: Is "methy-tag" a typographical error for "methyl tag?" If so, this is not a widely used term. This term should either be corrected, or introduced and defined.
We appreciate the reviewer’s comments. The "methy-tag" has been corrected as methyl-mark.
The paper has no figures or illustrations. A conceptual image illustrating the features they describe in the text would be very helpful.
We appreciate the reviewer’s comments. In revised manuscript, Figure 1 was added.
(x) Moderate English changes required
The manuscript has been edited by a native English speaker.
Reviewer 2 Report
Review of “Small Molecules Targeting the “Epigenetic-Tag” Readers in Cancer Therapy.” By H. Xhu, T. Wei, Y. Cai, and J. Jin.
This is a very nice review of the current state of what is known about epigenetic readers and small molecules targeting cancer. Although, I would prefer they expand it to include all small molecule work targeting these proteins.
Major:
The N-terminal tails of the nucleosome are not exposed in vivo, as they likely interact with the acidic patch on the neighboring nucleosomes to form nucleosomal arrays (line 25-26). In fact the whole first be expanded.
Table one is missing lots of references.
H3K23ac in not a “non-canonical histone signature” (line 117), in fact it is one of the most abundant (30-80% of all histones) to the fact that it can be difficult to measure changes by antibodies.
YEATS protein are also homologous to yAsf1 which has recently been shown to also recognize H3K14ac.
A more global discussion of how selectivity is achieved structurally would be helpful in understanding which ones might be more easily targeted by small molecules.
Another major discussion could also include how specificity was determined for each of these protiens.
Minor:
The title could be more specific they go in to a lot of what is known about the selectivity of these readers. Changing the title could expand the readership.
Author Response
Reviewer 2:Comments and Suggestions for Authors
This is a very nice review of the current state of what is known about epigenetic readers and small molecules targeting cancer. Although, I would prefer they expand it to include all small molecule work targeting these proteins.
Major:
The N-terminal tails of the nucleosome are not exposed in vivo, as they likely interact with the acidic patch on the neighboring nucleosomes to form nucleosomal arrays (line 25-26). In fact the whole first be expanded.We appreciate the reviewer’s comments. In new manuscript, the original sentence has been modified in accordance with the reviewers' comment.
Table one is missing lots of references.
References in Tables added.
H3K23ac in not a “non-canonical histone signature” (line 117), in fact it is one of the most abundant (30-80% of all histones) to the fact that it can be difficult to measure changes by antibodies.
We appreciate the reviewer’s comments. The original “non-canonical histone signature” was changed to "histone" in revised manuscript.
YEATS protein is also homologous to yAsf1 which has recently been shown to also recognize H3K14ac.
We added following sentence in section 2.1.3: (Interestingly, histone chaperone Asf1 in yeast is a structural homolog of the Yaf9, and recognizes H3K14ac. However, due to the lack of sequence similarity, it is not generally considered a YEATS protein [76]).
A more global discussion of how selectivity is achieved structurally would be helpful in understanding which ones might be more easily targeted by small molecules.
Another major discussion could also include how specificity was determined for each of these protiens.
In the revised manuscript, we added following sentences in related sections:
According to the structural features of BRDs, the important specificity determinants of BRD inhibitors are the so-called ‘WPF shelf ’and ‘gatekeeper’residue. The two residues of ‘WPF shelf’ change in hydrophobicity, size, or effects on backbone conformational properties, while ‘gatekeeper’ residue determins the shape of the entrance into the acetylation binding site [137] These differences make the two areas be explored to achieve selectivity depending on the desired targets.
Based on research data, there are two modes of epigenetic tags recognition of PHD fingers, the cavity insertion and surface groove [163], and these two modes are commonly seen in readers with PHD fingers and chromodomains, as well as MBT domains and the Tudor domain 53BP1, respectively [164]. Because of the high conservation of the binding areas, the usage of secondary pockets is the strategy of choice to achieve the specificity [165]
Minor:
The title could be more specific they go in to a lot of what is known about the selectivity of these readers. Changing the title could expand the readership.We appreciate the reviewer’s suggestions. The title has been changed to [Small Molecules Targeting the Specific Domains of Histone-Mark Readers in Cancer Therapy].
(x) English language and style are fine/minor spell check required
The manuscript has been edited by a native English speaker.
Reviewer 3 Report
This article reviewed protein domains that are involved in ‘epigenetic tags’ recognition and recent findings related to small molecules targeting ‘epigenetic tags’ -readers in cancer therapy.
In Section 3, a number of related clinical studies of small molecules targeting the epigenetic-tag readers in cancer therapy have been reviewed. However, these studies seem still in the clinically test stage. Are there any successful treatment or drug that has been developed based on this mechanism? Is the mechanism of small molecules targeting the epigenetic-tag readers related to epigenetics regulated by small non-coding RNAs?
Author Response
Reviewer 3:Comments and Suggestions for Authors
This article reviewed protein domains that are involved in ‘epigenetic tags’ recognition and recent findings related to small molecules targeting ‘epigenetic tags’ -readers in cancer therapy.
In Section 3, a number of related clinical studies of small molecules targeting the epigenetic-tag readers in cancer therapy have been reviewed. However, these studies seem still in the clinically test stage. Are there any successful treatment or drug that has been developed based on this mechanism? Is the mechanism of small molecules targeting the epigenetic-tag readers related to epigenetics regulated by small non-coding RNAs?To our knowledge, there have been several small molecules epigenetic drugs through the FDA approval such as DNMT1 inhibitors (Azacitidine, Decitabine), HDAC inhibitors (Vorinostat, Panobinostat, Romidepsin, Belinostat). However, so far, inhibitors targeting to epigenetic-tag readers are still in clinical trials or preclinical stage.
Also, the second question raised by the reviewer, we did not find relevant literature.
(x) English language and style are fine/minor spell check required
The manuscript has been edited by a native English speaker.
Round 2
Reviewer 1 Report
Authors have done an excellent job in preparing the revised manuscript. It is generally very much improved, and all of my concerns have been addressed.
I still detect some minor English issues in parts of the text. I believe it is most important that the abstract be correct, since it will advertise the paper. In particular, I would recommend to address the following issues in the abstract:
- the word "which" should be removed on line 19, and
- I would suggest to use "Here" rather than "Herein" on line 18; the construction may or may not be an error, but it stands out somewhat as an unusual stylistic choice.
The above comments are intended to be constructive, however, these do not affect the scientific evaluation. I believe the article is essentially ready for publication.
Reviewer 3 Report
This revised version is fine.